# Predictors of Treatment Requirements in Women with Gestational Diabetes: A Retrospective Analysis

**DOI:** 10.3390/jcm10194421

**Published:** 2021-09-27

**Authors:** Friederike Weschenfelder, Karolin Lohse, Thomas Lehmann, Ekkehard Schleußner, Tanja Groten

**Affiliations:** 1Department of Obstetrics, University Hospital Jena, 07747 Jena, Germany; friederike.weschenfelder@med.uni-jena.de (F.W.); ekkehard.schleussner@med.uni-jena.de (E.S.); 2Unit Neonatology, Department of Paediatrics, University Hospital Jena, 07747 Jena, Germany; karolin.lohse@med.uni-jena.de; 3Institute of Medical Statistics and Computer Science, University Hospital Jena, Friedrich Schiller University, 07747 Jena, Germany; thomas.lehmann@med.uni-jena.de

**Keywords:** gestational diabetes, insulin treatment, predictors, HbA1c, ultrasound parameters

## Abstract

The diagnosis of gestational diabetes is usually very stressful for pregnant women, especially because they fear that insulin treatment may become necessary. Knowledge about personal risk factors predicting the probability of insulin treatment could therefore help to improve acceptance of the diagnosis and therapy adherence. The aim of this study was to find potential risk factors for insulin dependency and treatment requirements using information available at the time of diagnosis of gestational diabetes during pregnancy. We included 454 singleton pregnancies diagnosed ≥24 weeks of gestation. Multivariate regression analysis was used to evaluate independent associations of metabolic, anthropometric and fetal ultrasound parameters with the general need for insulin treatment and further stratified treatment options: diet (*n* = 275), bolus insulin only (*n* = 45), basal insulin only (*n* = 73) and multiple daily injections (*n* = 61). Receiver operator characteristics and cut-off values for independent variables were generated. Treatment groups differed significantly concerning pre-pregnancy weight and BMI as well as fasting glucose and 1 h glucose test values. Significant cut-offs for insulin dependency were HbA1c level of 5.4%, FPG of 5.5 mmol/L and 1 h glucose of 10.6 mmol/L. At time of diagnosis, certain patient characteristics and measurements can help to predict treatment necessities and therefore improve individualized counselling.

## 1. Introduction

With a worldwide incidence of about 11%, gestational diabetes mellitus (GDM) is defined as a glucose tolerance disorder diagnosed during pregnancy following a standardized 75 g oral glucose tolerance test (OGTT) using IADPSG (International Association of the Diabetes and Pregnancy Study Group) criteria [1]. According to German S3 Guidelines [2], treatment is based on a structured training program after diagnosis, including dietary advice and training in self-measurement of blood glucose, as well as recommendations on physical activity and gestational weight gain. Women who are not able to achieve glucose targets by dietary intervention and physical activity require insulin treatment. According to German guidelines, metformin treatment is currently restricted to exceptional cases and not recommended. In Germany, about 30% of the GDM patients receive insulin treatment during pregnancy. Established treatment options are similar to those known in diabetes treatment. Short-acting insulin with meals only (‘bolus’), multiple daily injections of long and short acting insulin (‘MDI’) and injection of long acting (basal-)insulin only (‘basal’). The single injection of basal insulin at night means the least effort for the patients and hardly represents a burden for them.

Most patients fear the need of insulin therapy and experience the diagnosis of GDM as very distressing. Consistently, Draffin et al. showed that the need for insulin was often associated with fear, guilt and further concerns [3]. On the other hand, Mautner and Dorfer found that treatment with insulin had no negative effects on the emotional state of pregnant women [4], and Hussain et al. were able to show that patients with a more positive attitude and higher treatment satisfaction had numerically better glycemic control [5]. Thus, informed counseling of patients about their individual risk for insulin therapy at the time of diagnosis could therefore reduce the negative emotional burden associated with a diagnosis. Previous studies on predictors of the necessity of insulin treatment revealed maternal BMI, fasting plasma glucose (FPG) and 2 h blood glucose levels and HbA1c values to be associated with the requirement of insulin during pregnancy [6,7,8]. However, most of these studies compared diet control vs. insulin treatment, lacking further differentiation.

The aim of this study was to find predictors of both the general insulin requirement as well as for the described treatment subgroups within women diagnosed with GDM after 24 weeks of gestation. Our results could enable us to inform women about their individual risk for insulin dependency and expected treatment requirements at the time of diagnosis. This would help to avoid negative expectations and would facilitate individualized counseling and management of pregnant women diagnosed with GDM. Furthermore, risk assessment could constitute a profound release for those who do not have to expect insulin treatment. 

## 2. Materials and Methods

### 2.1. Study Population

A total of 990 singleton pregnancies with GDM were treated from 1 January 2012 until 31 December of 2017 in our outpatient unit for diabetes and pregnancy. Early-GDM cases (diagnosed < 24 weeks of gestation) (*n* = 131) and cases with incomplete data concerning the potential predictors (*n* = 405) were excluded from this study, and a total of 454 were included in the final analysis (Figure 1). The GDM diagnosis was based on IADPSG and WHO-2013 criteria [9,10]: fasting < 5.1 mmol/L; 1 h < 10 mmol/L; 2 h < 8.5 mmol/L at median 27 weeks of gestation (IQR 26–29). Diabetes care was provided according to the German S3 guidelines published in 2011 [11]. Ethical approval was given by the local Ethical Committee of the Friedrich-Schiller-University, Jena, Germany (5280-09/17).

### 2.2. Treatment Groups

Group comparisons of this study were based on four GDM treatment groups during pregnancy: diet (*n* = 275), bolus (*n* = 45), multiple daily injections (MDI) (*n* = 73) and basal (*n* = 61). All women received the same GDM training program following diagnosis, including dietary schooling, training in self-measurement of blood glucose (SMBG) as well as recommendations on physical activity and weight gain. Maternal glucose target levels were <5.3 mmol/L at fasting and <7.8 mmol/L at 1 h and <6.7 mmol/L at 2 h after eating. The diet subgroup achieved blood sugar goals with diet control and physical activity only and never required insulin. The bolus subgroup needed only mealtime insulin injections during pregnancy to achieve postprandial blood sugar control. According to our internal guidelines, we start with fast-acting human insulin, using rapid-acting insulin analogue only second line if postprandial peaks remain. The MDI subgroup required long- and short-acting insulin in a multiple daily injection (MDI) regime, combining short- and long-acting insulin. For the basal insulin at night, intermediate-acting NPH insulin was used primarily, and insulin detemir as a long-acting insulin analogue was used second line in cases in which fasting glucose targets could not be achieved. The basal subgroup needed only a single dose of basal insulin at night to reach glucose targets. All women received personal education on insulin and injection procedures. Women with insulin were seen fortnightly, and the diet subgroup was seen every four weeks. Women were stratified according to the maximum therapy needed during pregnancy. None of the women were additionally treated with metformin. 

### 2.3. Data Collection

Patient characteristics, history and family history were retrieved from patient records. BMI was calculated using maternal height and the documented prepregnancy weight and grouped according to the definitions of the World Health Organization (WHO) [12]. Maternal blood pressure and HbA1c levels were taken at the time of diagnosis. Fetal biometry (including abdominal circumference (AC) and estimated fetal weight (EFW) was performed using standardized anatomic views according to ISUOG guidelines [13] (see Table 1). 

### 2.4. Statistical Analysis

Statistical analysis was performed with SPSS 24.0. We included all patients with inclusion and without exclusion criteria in our analysis. Chi^2^ test or Fisher exact test were used to compare categorical data. Continuous data are presented using median and interquartile range (IQR) due to the data lacking a normal distribution. We used the Wilcoxon test to compare the four subgroups: diet, bolus, MDI and basal. The Bonferroni correction was used due to multiple testing. A multinomial logistic regression was performed to model the associations between the predicting factors and insulin treatment as well as the treatment subgroups (bolus, basal and MDI). In order to get unbiased estimates in the multivariate logistic regression model, a prior sample size estimation was performed for the primary outcome variable—insulin treatment. The minimum number of necessary events per included variable were observed in the cohort (*n* = 179 women with insulin treatment), which was sufficient for the analyses of the 17 independent variables included in our multivariate analysis [14]. Included variables were maternal age (years), gestational age at diagnosis, parity, prepregnancy weight (kg), prepregnancy BMI (kg/m^2^), systolic and diastolic blood pressure (mmHg), history of GDM, thyroid disorders, cardiovascular disorders, family history of diabetes, HbA1c at diagnosis (%) and fasting plasma glucose (FPG), 1 h plasma glucose and 2 h plasma glucose (all in mmol/L). Odds ratios (ORs) with 95% confidence intervals (CI) are presented. Receiver operator characteristics (ROC) analyses were performed to evaluate the accuracy of the prediction of the certain treatment methods by using AUC with 95% confidence intervals. Cut-off regarding the prediction model was determined by Youden index criteria, and sensitivity and specificity are reported [15] and used for calculating the negative and positive predictive values (NPV and PPV) (see Table 3). A *p*-value < 0.05 was considered to indicate statistical significance (2-tailed).

## 3. Results

### 3.1. Baseline Characteristics of Subgroups 

Subgroup characteristics at time of GDM diagnosis (27 weeks of gestation (IQR 26; 29) are shown in Table 1. The subgroups did not differ concerning maternal age, gestational age at diagnosis, parity, blood pressure values, family history of diabetes, thyroid disorders, cardiovascular disorders, smoking, EFW percentile at diagnosis and 2 h blood glucose values in the 75 g oGTT.

Women requiring insulin treatment were significantly heavier (78 kg; IQR 68–92 vs. 66 kg; IQR 58–77) (*p* < 0.01) and had higher BMI level (27.8 kg/m^2^; IQR 24.4–33.5 vs. 24.2 kg/m^2^; IQR 21.7–28.0) (*p* < 0.1) compared with diet controlled. Overall obesity rate was 40.4% in the insulin dependent and 16.1% in the insulin independent diet group (*p* < 0.01).

Women in the treatment subgroup MDI showed the highest prepregnancy weight (80 kg; IQR 70–95) followed by those in the bolus (76 kg; IQR 67–88) and basal (74 kg; IQR 68–91) subgroups. Accordingly, prepregnancy BMI was the highest in the MDI subgroup (29 kg/m^2^; IQR 24.8–34.5), followed by bolus (27.9 kg/m^2^; IQR 23.5–33.2) and basal (26.4 kg/m^2^; IQR 24.6–32.7) subgroups (*p* < 0.01). Looking at the BMI categories, significantly more obese GDM mothers were found in the MDI (45.7%) and bolus (40%) subgroup compared with the basal subgroup (32.6%).

Fetal ultrasound measurements at time of diagnosis such as the fetal AC and median fetal weight were within normal ranges, but there were still significant group differences: AC percentiles 68 (IQR 46–83) vs. 53 (IQR 34–71) and EFW percentiles 52 (IQR: 29–79) vs. 39 (IQR: 23–63) were significantly higher in the insulin group compared with the diet group (*p* < 0.001). 

Accordingly, insulin subgroups showed significant higher AC and EFW percentiles in the MDI group (74; IQR 50–86; 60; IQR 30–85), followed by the basal (68; IQR 49–81; 60 IQR 31–84; 77) and bolus (55; IQR 41–76; 42 IQR 26–63) subgroups (*p* < 0.001). 

HbA1c levels at diagnosis were significantly higher in the insulin-treated group (5.4%; IQR 5.1–5.6) compared with the diet group (5.2%; IQR 4.9–5.4) and differed significantly within the insulin treatment subgroups, showing the highest value in the MDI group (5.5%; 37 mmol/mol), followed by bolus subgroup with 5.4% (36 mmol/mol) and basal subgroup with 5.3% (34 mmol/mol). The 75 g oGTT results differed significantly at fasting and 1 h, with the highest FPG in the MDI subgroup (5.6 mmol/L; IQR 5.2–6.1), followed by the basal (5.3 mmol/L; IQR 5.1; 5.6) and then the diet and bolus subgroups, with 5.1 mmol/L as the mean FPG (*p* < 0.01). Lowest 1 h glucose levels were seen in the diet subgroup with 9.6 mmol/L, followed by basal (10 mmol/L) and 10.6 mmol/L both in the bolus and the MDI group (*p* < 0.01). The 2 h values did not show relevant differences. 

### 3.2. Factors Associated with Insulin Treatment and Treatment Subgroups

Multivariate analysis revealed that different independent variables were statistically significant (see Table 2). The risk to develop insulin requirement was significantly increased with FPG (OR 1.65; CI 1.14–2.39), 1 h glucose (OR 1.24; CI 1.07–1.44) and HbA1c (%; OR 3.99; CI 1.99–8.05). Gestational age at diagnosis and diastolic blood pressure were inversely associated with insulin treatment: GA (in weeks, OR 0.82; CI 0.74–0.92) and diastolic blood pressure (mmHg; OR 0.96; CI 0.93–0.99). Bolus insulin during pregnancy was significantly affected by 1 h glucose (mmol/L; OR 1.47; CI 1.17–1.87) and HbA1c (%, OR 4.08; CI 1.43–11.65). The necessity for basal insulin only was independently affected by FPG (OR 1.86; CI 1.12–3.08) and a history of GDM (OR 3.25; CI 1.20–8.83). MDI therapy was associated with FPG (OR 2.38; CI 1.42–3.97), 1 h glucose (OR1.35; CI 1.09–1.67), HbA1c (%, OR 9.79, CI 3.58–26.78), AC percentile (OR 1.046; CI 1.02–1.07) and diastolic blood pressure (OR 0.95, CI 0.90–0.99).

### 3.3. Cut-Off Values for Predictive Independent Variables for Treatment Subgroups

For the identified significant independent variables, area under the curve (AUCs), cut-off values, sensitivity (Sens.), specificity (Spec.), negative predictive values (NPV) and positive predictive values (PPV) were calculated (see Table 3). The cut-off values for insulin treatment were 5.5 mmol/L for FPG (Sens. 42.5, Spec. 84.4, AUC 0.643), 10.6 mmol/L for 1 h glucose (Sens. 45.3, Spec. 76.7, AUC 0.643) and 5.4% for HbA1c (Sens. 66.9, Spec. 54.7, AUC 0.653); NPV and PPV are shown in Table 3. For bolus treatment, cut-off values were 10.6 mmol/L for the 1 h glucose (Sens. 51.1, Spec. 70.2; AUC 0.630) and 5.4% for HbA1c (Sens. 57.8, Spec. 60.1, AUC 0.600)—both with a high NPV of 92.9% and 92.8%, respectively. Cut-off value for basal treatment was 5.2 mmol/L for FGP (Sens. 70.5, Spec. 48.9; AUC 0.613), with a NPV of 91.4%. MDI treatment cut-off values were 5.6 mmol/L for the FPG (Sens. 56.2, Spec. 83.7; AUC 0.723), 10.7 mmol/L for the 1 h glucose (Sens. 49.3, spec. 73.5; AUC 0.655), 5.4% for the HbA1c (Sens. 69.9; Spec. 63.84; AUC 0.734) and 69th percentile of AC (Sens. 60.3; Spec. 67.2; AUC 0.662); NPV and PPV were very variable concerning MDI and are shown in Table 3.

## 4. Discussion

Our study revealed HbA1c and fasting and 1 h glucose in the 75 g oGTT to be of significant predictive value for identifying individuals who will most likely not require insulin treatment during their GDM pregnancies. With a cut-off value of 5.5 mmol/L for fasting glucose, 10.6 mmol/L for 1 h glucose and HbA1c values below 5.4%, we can rule out the need of insulin therapy with a NPV of 69, 68 and 69, respectively. Looking at the different treatment groups, NPV increases. The need of basal insulin therapy can be ruled out with a NPV of 91 using a cut-off of 5.2 mmol/L for fasting glucose. For bolus insulin therapy, the cut-offs were 10.6 mmol/L for the 1 h glucose (NPV 93) and 5.4% for HbA1c (NPV 93). NPV for MDI therapy was 91 for a cut-off of 5.6 mmol/L for fasting glucose, 88 for a cut-off of 10.7 mmol/L for 1 h glucose, 92 for a Hb1Ac below 5.4% and 90 for fetal abdominal circumference below the 69th percentile (Table 3).

Although, NPV were high, PPV and sensitivity were low; thus, the described risk factors were revealed to be valuable for predicting the absence of insulin necessity rather than for predicting insulin requirement. This constitutes a particular relief for those meeting cut-off criteria and may motivate them to stick to advice on diet and physical activity to successfully avoid insulin treatment. Thus, the cut-offs described provide new information that might be included in the primary counselling of GDM patients and might contribute to their motivation to adhere to treatment. 

NPVs for the cut-off values revealed for any insulin treatment were lower compared with those for the different treatment options, further emphasizing the diverse characteristics of the here-described groups. Accordingly, the here-presented comparison of group characteristics profoundly confirmed the diversity of the group of GDM patients (Table 1).

Most of the previous published studies on GDM-related topics so far have not differentiated treatment approaches regarding insulin administration. Individualized patient-centered management as carried out at our department resulted in the four heterogeneous groups described here. The diet group was sufficiently controlled by dietary changes and physical activity. The bolus group solely required short-acting insulin to control postprandial glucose levels. The MDI group required classical insulin treatment including long- and short-acting insulin and individuals in the basal group showed constantly elevated fasting glucose levels requiring a single dose of long acting basal insulin at night. All groups revealed equally sufficient glucose control and perinatal outcome (see Appendix A). However, we did see significant differences in group characteristics regarding prepregnancy maternal weight and BMI. The MDI group as the most insulin dependent group was revealed to present with the highest values for median prepregnancy weight (80 kg) and prepregnancy BMI (29 kg/m^2^) and the highest frequency of obese women (45.7%). However, BMI did not appear to be of independent predictive value concerning treatment requirements. Even more surprisingly, a history of GDM in previous pregnancies did not appear to impact individually on therapeutic requirements.

Comparing the results of the 75 g oGTT, we observed significant differences at fasting and 1 h glucose levels between groups. As expected, the MDI group had the highest overall values at all three time-points. MDI and basal subgroups showed the highest fasting values (5.6 mmol/L and 5.3 mmol/L; *p* < 0.01) compared with diet and bolus, with 5.1 mmol/L, thus revealing a risk for continuous elevated overnight glucose levels. Interestingly, the basal subgroup showed high fasting values while presenting the second lowest 1 h and the lowest 2 h glucose levels. This phenomenon seems to be particularly characteristic of this special subclass. Law et al. published CGM-based glucose analysis of GDM pregnancies and showed that mothers of LGA infants ran significantly higher glucose overnight compared with mothers without LGA infants. Consequently, the authors recommended detection and improved control of nocturnal glucose levels to prevent fetal overnutrition [16]. Accordingly, looking at fetal development at the time of diagnosis in our group we could already see differences, with the highest median percentiles of abdominal circumference in the MDI (74th percentile) and basal (68th percentile) subgroups. Thus, both subgroups with need for basal insulin already presented with higher AC at the time of diagnosis. This may reflect the effect of elevated nocturnal glucose levels on fetal development, as supposed by Law et al. [16], and supports the higher probability of insulin requirement in this subgroup. Multivariate analysis showed that fetal AC percentile was independently associated with MDI (OR 1.046). Thus, our results emphasize clearly that elevated levels of fasting glucose should gain more attention in the management of GDM. 

Previous studies on predictors for insulin therapy during pregnancy focused on oGTT values and maternal anthropometric measurements only. In concordance to our data, they showed that FPG levels and HbA1c at diagnosis were highly associated with insulin treatment during pregnancy [6,7,8,17]. In contrast with our data, some authors showed that 2 h blood glucose and maternal BMI were independent predictors [6,8] of insulin requirement. Since two of these studies of Zhang et al. and Tang et al. were performed on a group of Asian GDM patients [7,8], results might not be transferable due to ethnic differences. Furthermore, Yien et al. highlight the fact that clinicians should be mindful of the impact that differences in ethnicity may have on outcome and management [18]. Wong et al. showed that elevated BMI was related to insulin therapy in an Australian cohort, again with a high ethnic diversity, but with a very high percentage of insulin treated patients (52.8%) [19], reducing comparability with our data. Our cohort was especially homogenous, with more than 90% Caucasians. It also needs to be mentioned that glucose thresholds applied in the study of Wong et al. were higher, with FPG of 5.5 mmol/L and 2 h blood glucose target of 7.0 mmol/L. Additionally, compared with our study, they used categorical variables for BMI, which might be another reason for our different findings [6]. 

In contrast with BMI, HbA1c levels remained an individual predictor after multivariate analysis for insulin, bolus and MDI only but not for basal. As HbA1c levels in general show a higher correlation with postprandial glucose values rather than FPG levels, this might be an explanation for this exception [20]. Thus, in women with selectively elevated glucose values at night, HbA1c levels might not be as helpful for evaluating blood glucose control. Nevertheless, at diagnosis, HbA1c levels above 5.3% were associated with an increased risk for insulin treatment in Tang et al. [7] and also in our study. HbA1c above 5.4% was associated with the need for bolus insulin (bolus and MDI group) with a very high NPV 93 and 92, respectively. 

To our knowledge, there are no studies so far that differentiate between the heterogeneous treatment options when investigating the predictability of insulin requirement. Our results show that using easily accessible data on maternal anthropometrics, patient’s history, general laboratory results and ultrasound parameters might help to differentiate between future treatment approaches at the time of diagnosis and thus could lead to straight forward and more consequential management.

Being diagnosed with GDM during pregnancy, an already stressful time during a women’s life, seems to be an extra stressor—especially if insulin treatment becomes inevitable [3]. Knowledge about individual risks for the need of insulin treatment gives time to cope and will enhance motivation to adhere to management advice, especially since there is evidence that positive maternal attitude and satisfaction can help improve blood sugar values [5]. Interestingly, known risk factors for GDM such as BMI, maternal age and previous GDM did not interfere with an insulin requirement. Therefore, parameters collected at the time of diagnosis such as 75 g oGTT values, HbA1c and fetal ultrasound parameters have to be considered and discussed with the patient. 

### Limitations

A strength of this study was the rather large number of complete data on patients diagnosed with GDM after 24 weeks of gestation, including patient’s history, body measurements, laboratory results and ultrasound parameters. There are certain limitations of the present study that include the retrospective design, the reliability of the electronic records and the unicentric approach. Since we decided to include a large number of variables in our regression model, the number of cases that were excluded from the analysis, caused by mostly one missing value, was particularly large. However, characteristics of the initial entire cohort of 990 cases are comparable with this study’s final cohort (maternal age 31 years, maternal BMI 26, 5 kg/m^2^, insulin therapy 42.2%, bolus 11.6%; basal 15.3%; MDI 15.5%, HbA1c 5.2% at diagnosis (data not shown). Prior sample size estimation for the multivariate analysis focused on the primary outcome variable—insulin treatment only. The minimum number of necessary events per included variable was observed [14]. 

Another limitation that needs to be mentioned is the relatively high percentage of insulin-treated pregnancies (39.4%) due to frequent admission of high-risk GDM cases to our specialized unit. The general rate of insulin treatment in Germany is about 30% [21]. Nevertheless, the rate of insulin-treated patients in our cohort was still lower than in the mentioned studies of Wong, with 52.8%. However, blood glucose targets and indications for insulin treatment vary between countries and guidelines followed, which limits the comparability of these data and represents a limitation in itself. Another limitation is that we could not consider further possible predictors (e.g., HOMA Index, insulin levels) for our multivariate analysis. Outcome data were not available for all patients, as a further limitation. However, perinatal outcome did not differ between the treatment groups, which we think might reflect successful treatment and management keeping glucose levels within targets. With regard to ethnicity, it should be noted that in the obstetric population of our department, the proportion of non-Caucasian women is well below 10%, and therefore our results may be representative only for Caucasian GDM patients. Finally, we did not account for socioeconomic factors in our study, which might constitute an additional risk for developing insulin dependency during pregnancy due to reduced capability to adhere to dietary and physical activity advice.

## 5. Conclusions

This study confirmed that women being diagnosed with GDM show a wide range of characteristics. Despite this heterogeneity, HbA1c and fasting and 1 h glucose in the 75 g oGTT were of significant predictive value for identifying individuals who would most likely not require any insulin treatment during their GDM pregnancies. However, cut-off values were different in treatment subgroups. A cut-off of 5.4% for HbA1c ruled out insulin necessity for all individuals requiring bolus insulin (MDI and bolus only), while fasting glucose values below 5.2 mmol/L ruled out the need for bolus insulin only with a high NPV. Thus, our study will help to develop individualized counseling and management approaches that significantly improve patient care and increase emotional comfort.

## Figures and Tables

**Figure 1 jcm-10-04421-f001:**
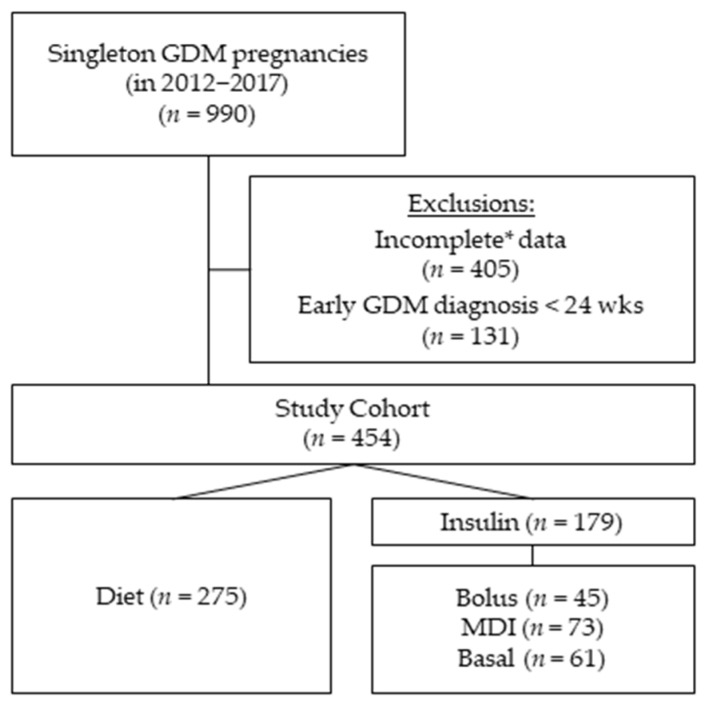
Cohort composition: The study cohort consists of 454 women diagnosed with gestational diabetes (GDM) at 24 weeks of gestation or later. Of the initial 990 women treated with GDM in our outpatient clinic having singleton pregnancies between 2012 and 2017. We excluded all women with GDM diagnosis before 24 weeks of gestation (*n* = 131) and cases with * incomplete variables needed for the regression multivariate analysis (*n* = 405). Group comparisons of this study were based on four GDM treatment groups during pregnancy: diet (*n* = 275), bolus (*n* = 45), multiple daily injections (MDI) (*n* = 73) and basal (*n* = 61).

**Table 1 jcm-10-04421-t001:** Predictive parameters for treatment groups based on characteristics.

Variable	Total Cohort (*n* = 454)	Diet Control (*n* = 275)	Insulin (*n* = 179)	*p* ^†^	Bolus (*n* = 45)	Basal (*n* = 61)	MDI (*n* = 73)	*p* ^‡^
Age in years	31 (27–34)	31 (27–34)	31 (28–34)	0.630	32 (27–35)	31 (28–33)	31 (28–34)	0.995
Parity	1(0–1)	0 (0–1)	1 (0–1)	0.013	1(0–1)	1(0–1)	1(0–2)	0.083
Pre-pregnancy weight in kg	70 (60–85)	66 (58–77)	78 (68–92)	<0.001 *	76 (67–88)	74 (68–91)	80(70–95)	<0.001 *
Pre-pregnancy BMI in kg/m²	25.4 (22.5–30.1)	24.2 (21.7–28.0)	27.8 (24.4–33.5)	<0.001 *	27.9 (23.5–33.2)	26.4 (24.6–32.7)	29 (24.8–34.5)	<0.001 *
Pre-pregnancy BMI subgroups				<0.001 *				<0.001 *
Underweight (<18.5 kg/m^2^)	1.1%	1.8%	0		0%	0%	0%	
Normal weight (18.5–24.9 kg/m^2^)	45.5%	55.3%	32.3%		37.8%	39.1%	24.3%	
Overweight (25–29.9 kg/m^2^)	27%	26.7%	27.3%		22.2%	28.3%	30%	
Obesity (≥30 kg/m^2^)	26.5%	16.1%	40.4%		40%	32.6%	45.7%	
Systolic BP	122 (114–131)	120 (112–130)	125 (120–134)	0.019	120 (110–133)	126 (120–134)	125 (120–135)	0.037
Diastolic BP	70(70–85)	80 (70–85)	80 (70–85)	0.967	80 (70–85)	80 (71–84)	78 (70–85)	0.927
History of GDM	7.7%	6.5%	9.5%	0.282	6.7%	14.8%	6.8%	0.178
Thyroid disorders	17%	13.5%	22.3%	0.015	17.8%	23%	24.7%	0.07
Cardiovascular disorders	10.4%	9.8%	11.2%	0.640	17.8%	11.5%	6.8%	0.283
Family history of diabetes	52.6%	50.9%	55.3%	0.387	60%	49.2%	57.5%	0.512
Smoking	19.6%	20.7%	17.9%	0.471	11.1%	19.7%	20.5%	0.508
Percentile of EFW	43 (24–69)	39 (23–63)	52 (29–79)	<0.001 *	42 (26–63)	60 (31–84)	60 (30–85)	<0.001 *
Percentile of AC	58 (38–76)	53 (34–71)	68 (46–83)	<0.001 *	55 (41–76)	68 (49–81)	74 (50–86)	<0.001 *
HbA1c at diagnosis in %	5.3 (5.0–5.5)	5.2 (4.9–5.4)	5.4 (5.1–5.6)	<0.001 *	5.4 (5.1–5.5)	5.3 (5.1–5.4)	5.5 (5.3–5.8)	<0.001 *
Gestational age at diagnosis (weeks)	27 (26–29)	28 (26–29)	27 (26–29)	0.006	27 (25–29)	27 (26–29)	27 (26–29)	0.011
75 g oGTT in mmol/L FPG	5.2 (4.8–5.5)	5.1 (4.7–5.3)	5.3 (5.0–5.8)	<0.001 *	5.1 (4.5–5.5)	5.3 (5.1–5.6)	5.6 (5.2–6.1)	<0.001 *
1 h blood glucose	9.9 (8.5–10.8)	9.6 (8.2–10.5)	10.3 (9.1–11.2)	<0.001 *	10.6 (9.4–11.4)	10 (8.4–10.7)	10.6 (9.3–12.2)	<0.001 *
2 h blood glucose	7.7 (6.5–8.9)	7.7 (6.4–8.8)	7.8 (6.7–9.1)	0.139	7.5 (6.3–9.0)	7.4 (6.7–8.5)	8.1 (6.9–10)	0.045

* Significant after Bonferroni correction for multiple testing (*p* < 0.05); ^†^ Comparing diet vs. insulin groups. ^‡^ Comparing diet, bolus, basal and MDI (multiple daily injections); AC, abdominal circumference; BMI, body mass index; BP, blood pressure; EFW, estimated fetal weight; FPG, fasting plasma glucose; GDM, gestational diabetes; FPG, fasting plasma.

**Table 2 jcm-10-04421-t002:** Multivariate logistic regression analysis.

Treatment Group	Independent Variable	OR	CI	*p*
Insulin	FPG in mmol/L	1.65	(1.14–2.39)	<0.01
	1 h glucose in mmol/L	1.24	(1.07–1.44)	<0.01
	HbA1c at diagnosis in %	3.99	(1.99–8.05)	<0.01
	GA at diagnosis	0.82	(0.74–0.92)	<0.01
	Diastolic blood pressure in mmHg	0.96	(0.93–0.99)	0.02
Bolus	1 h glucose	1.47	(1.17–1.87)	<0.01
	HbA1c at diagnosis in %	4.08	(1.43–11.65)	<0.01
Basal	FPG in mmol/L	1.86	(1.12–3.08)	0.02
	History of GDM	3.25	(1.20–8.83)	0.02
MDI	FPG in mmol/L	2.38	(1.42–3.97)	<0.01
	1 h glucose in mmol/L	1.35	(1.09–1.67)	<0.01
	HbA1c at diagnosis in %	9.79	(3.58–26.78)	<0.01
	AC percentile at diagnosis	1.046	(1.02–1.07)	<0.01
	Diastolic blood pressure in mmHg	0.95	(0.90–0.99)	0.02

AC, abdominal circumference; FPG, fasting plasma glucose; GDM, gestational diabetes; MDI multiple daily injections; OR, odds ratio. Included parameters in logistic regression: maternal age (years), parity, prepregnancy weight (kg), prepregnancy BMI (kg/m^2^), systolic and diastolic blood pressure (mmHg), history of GDM, thyroid disorders, cardiovascular disorders, family history of diabetes, HbA1c at diagnosis (%) and fasting plasma glucose (FPG), 1 h plasma glucose, 2 h plasma glucose, gestational age at diagnosis, estimated fetal weight percentile and abdominal circumference percentile. Only significant ORs are presented (*p* < 0.05).

**Table 3 jcm-10-04421-t003:** Presentation of cut-off values, sensitivity, specificity, negative predictive value and positive predictive value of independent variables.

Treatment Group	Independent Variable	AUC	Cut-Off	Sens.	Spec.	NPV	PPV
Insulin	FPG (mmol/L)	0.643 (0.590–0.696)	5.5	42.5	84.4	69.3	63.9
	1 h glucose (mmol/L)	0.643 (0.582–0.686)	10.6	45.3	76.7	68.3	55.9
	HbA1c (%)	0.653 (0.603–0.704)	5.4	66.9	54.7	69.4	51.9
Bolus	1 h glucose (mmol/L)	0.63 (0.55–0.711)	10.6	51.1	70.2	92.9	15.9
	HbA1c (%)	0.60 (0.525–0.675)	5.4	57.8	60.1	92.8	13.8
Basal	FPG (mmol/L)	0.613 (0.543–0.682)	5.2	70.5	48.9	91.4	17.6
MDI	FPG (mmol/L)	0.723 (0.653–0.793)	5.6	56.2	83.7	90.9	39.8
	1 h glucose (mmol/L)	0.655(0.583–0.727)	10.7	49.3	73.5	88.3	26.3
	HbA1c (%)	0.734 (0.672–0.796)	5.4	69.9	63.8	91.7	27
	AC percentile	0.662 (0.591–0.733)	69	60.3	67.2	89.8	26

AC, abdominal circumference; FPG, fasting plasma glucose; GDM, gestational diabetes; Sens., Sensitivity; Spec., Specificity; NPV, negative predictive value; PPV, positive predictive value. Only significant AUCs are presented in the table (*p* < 0.05).

## Data Availability

The data presented in this study are available on reasonable request from the corresponding author.

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
