# Peer review of "Predictors of Treatment Requirements in Women with Gestational Diabetes: A Retrospective Analysis"

_jcm, 2021, doi:10.3390/jcm10194421_

Round 1

Reviewer 1 Report

General review:

Overall comments for the manuscript:

This study (a retrospective analysis) aimed to identify individual risk factors/predictors for both, a general insulin requirement and for the described therapeutic subgroups within women diagnosed with gestational diabetes mellitus (GDM) after at 24 weeks of gestation or later. The diagnosis of GDM is distressing for pregnant women, and with a worldwide incidence being up to 11 % it is an important study area - not just for improving the overall health of the pregnant mothers and managing the pregnancy outcomes - but studies in this area will help to improve the health of their offspring too. In total 487 singleton pregnancies that were diagnosed at ≥24 weeks of gestation in the outpatient department for diabetes and pregnancy at University Hospital Jena (Germany) (after being consulted from 1st of January 2012 until 31st of December 2017), were included in the study. Group comparisons of this study were based on four GDM treatment groups (N=454) during pregnancy: ‘Diet’ (n=275), ‘Bolus’ (n=45), ‘multiple daily injections (MDI)’ (n=73) and ‘Basal’ (n=61). A multivariate regression analysis was applied to evaluate independent associations of metabolic, anthropometric and fetal ultrasound parameters with the general need for insulin and the different treatment options (diet, bolus insulin only, basal insulin only and multiple daily injections). Authors concluded that different cut-offs for the requirement of insulin treatment (HbA1c level of 5.4%, FPG of 5.5 mmol/l and 1h glucose of 10.6 mmol/l) could be identified for the treatment subgroups (treatment groups differed significantly at time of diagnosis concerning BMI, pre-pregnancy weight, percentiles of estimated fetal weight and abdominal circumference, fasting glucose and 1hour glucose of 75g oral glucose tolerance test), and that patient characteristics can predict the requirement for the treatment. Authors state that these results enable physicians to inform women about their individual risk for insulin dependency and expected treatment requirements at the time of diagnosis.

Comments for Title and Abstract:

  1. A general suggestion: I advise the authors to rewrite the Abstract in a way so that the relevant parts from the article are pointed out to really capture the realty important message of the present paper – think of writing the abstract by providing the relevant parts I right order (example 1 on lines 11-13: seem to be those that would normally be in the end of the abstract). In its current state, it is not serving the rest of the manuscript as good as it could. Keep the focus, and provide only what is in the focus.

  1. I advise the authors the rephrase the lines 11-12 (in current state it is way too packed and message is not clear): “Knowledge about individual risk factors for expected treatment requirements as the need of insulin at the time of diagnosis”.

  1. On lines 16-17: This is a good place to tell really clearly about the grouping (“study arms; 4 + the 33”) more clearly than it is now (see a comment below about this).

  1. What is the take home message based on this study? It has been stated more clear in other parts of the paper, yet it is really important to have in Abstract as well in a really clear format.

Comments for Introduction:

  1. Introduction could use a bit more clear “red line” – I would advice the authors to cut some text that is taking the reader more far from the focus and to make it a more clear why and how the mental aspects of adhering to the insulin treatment is discussed, and how it supports the current study design where 5 groups are compared (4 + the 33, see a comment below)? Also, what are the general factors that predispose to GDM?

  1. A brief suggestion: I advise the authors to add a short but relevant sentence, where the risk of GDM in relation to pre-pregnancy diabetes is discussed just to add another angle to the story.

Comments for the other parts of the manuscript:

  1. I would ask the authors to discuss the results in the relation to pre-pregnancy diabetes a bit more in Discussion (to connect with the suggested sentence in Introduction).
  2. How was the final study population defined, is this full exclusion and inclusion criteria?
    1. Were there any pregnancies, where mothers have been previously diagnosed with any of the other diabetes sybtypes (T2D, T1D etc), and if yes, how were those individuals treated in this analysis (excluded/included?)?
    2. In case they were included, could an additional sensitivity analysis been done where those pregnancies are excluded?
    3. Were there differences among first-time pregnancies and others? If not done, a quick sensitivity analysis could be done.
    4. Were there pregnancies, where a mother has been also previously diagnosed with GDM at any of the previous pregnancies?
    5. Were there differences between the different age groups? How much could age explain of these/potential differences related to treatment? I would advise the authors to discuss this more in Discussion.
    6. Were any of the initial 990 mother-child pairs excluded due to prior metforming treatment? Was this also an exclusion criteria as it seems to have been stated as such on line 96?
  3. In this study, 487 mother-offspring pairs are included, what is the power? I would kindly ask the authors to perform the power calculations and add them as a supplementary table, if possible.
  4. I would kindly ask the authors to provide more information to the Figure 1, by additional boxes for treatment groups (this will help the reader to get the message faster), where this part of the text is visualized: both, a general insulin requirement and for the described therapeutic subgroups(Group comparisons of this study were based on four GDM treatment groups during pregnancy: ‘Diet’ (n=275), ‘Bolus’ (n=45), ‘multiple daily injections (MDI)’ (n=73) and ‘Basal’ (n=61)) within women diagnosed with gestational diabetes mellitus (GDM) after at 24 weeks of gestation or later”.
  5. Total study population is 487, and the N:s for four GDM treatment groups during pregnancy: ‘Diet’ (n=275), ‘Bolus’ (n=45), ‘multiple daily injections (MDI)’ (n=73) and ‘Basal’ (n=61) = 454. What is with the remaining 33 mother-child pairs? Are they a fifth study arm referred to as “a general insulin requirement”-group? I advice the authors to mention the “study arms” briefly in 2.1 Study Population, add “all 5 =4 treatment groups plus those 33 pairs?” groups to Figure 1, and mention them already in the Abstract to make this more clear to the readers from very beginning.
  6. In 2.3 Data collection, I would ask the authors to provide more information on population characteristics in a Table (Since Table 1 already exists, it would be great to add the reference to it already here).
  7. In Table 1, it is stated that total cohort N=454, however in Figure 1 and abstract, and some other parts in the text the number is 487. As advices above, make sure to explain what happened to the 33 pairs clearly from the beginning and to modify the Figure 1 accordingly. This will make it clear to the future readers as well.
  8. I can see that ROC analyses were performed to evaluate the accuracy of the prediction of the certain treatment methods by using AUC with 95% confidence intervals. I would ask the authors to generate a supplementary table with the relevant AUC(ROC) results, and add a reference to the relevant Supplementary Table to line 122? In addition, were all variables modelled as continuous variables?
  9. In 2.4, were the analysis models adjusted for which factors? Add all relevant information to the results and other relevant parts.
  10. Could these risk factors be utilized for GDM-specific risk score generation? Further, how about the score, could it be used for disease prediction? An assessment of predictive performance of GDM-risk score for GDM-diagnosis would be of clinical and epidemiological relevance. This interesting topic could be briefly discussed in Discussion (with relevant references).
  11. Could some of these observed differences been explained by socioeconomic or environmental differences? This could be further discussed.
  12. I would advice the authors to start Discussion with summing up their own aim and results (keep the general rules about the chapter order in mind here as well) instead of the current first sentence.

Other minor comments:

  1. Overall, readability is good, but in some places it could be improved.

Author Response

Dear Editor,

We would like to express our explicit thank the reviewers for careful reading our paper and providing such valuable comments. We are particular happy to respond to the comments provided and are convinced the manuscript has improved substantially.

Please find our point to point response to reviewer 1 on the following pages.

We deeply hope you will find the paper now acceptable to be published in your Journal.

Sincerely Tanja Groten

Response to Reviewer 1  

General review:

Overall comments for the manuscript:

This study (a retrospective analysis) aimed to identify individual risk factors/predictors for both, a general insulin requirement and for the described therapeutic subgroups within women diagnosed with gestational diabetes mellitus (GDM) after at 24 weeks of gestation or later. The diagnosis of GDM is distressing for pregnant women, and with a worldwide incidence being up to 11 % it is an important study area - not just for improving the overall health of the pregnant mothers and managing the pregnancy outcomes - but studies in this area will help to improve the health of their offspring too. In total 487 singleton pregnancies that were diagnosed at ≥24 weeks of gestation in the outpatient department for diabetes and pregnancy at University Hospital Jena (Germany) (after being consulted from 1st of January 2012 until 31st of December 2017), were included in the study. Group comparisons of this study were based on four GDM treatment groups (N=454) during pregnancy: ‘Diet’ (n=275), ‘Bolus’ (n=45), ‘multiple daily injections (MDI)’ (n=73) and ‘Basal’ (n=61). A multivariate regression analysis was applied to evaluate independent associations of metabolic, anthropometric and fetal ultrasound parameters with the general need for insulin and the different treatment options (diet, bolus insulin only, basal insulin only and multiple daily injections). Authors concluded that different cut-offs for the requirement of insulin treatment (HbA1c level of 5.4%, FPG of 5.5 mmol/l and 1h glucose of 10.6 mmol/l) could be identified for the treatment subgroups (treatment groups differed significantly at time of diagnosis concerning BMI, pre-pregnancy weight, percentiles of estimated fetal weight and abdominal circumference, fasting glucose and 1hour glucose of 75g oral glucose tolerance test), and that patient characteristics can predict the requirement for the treatment. Authors state that these results enable physicians to inform women about their individual risk for insulin dependency and expected treatment requirements at the time of diagnosis.

We thank the reviewer for acknowledging or work.

Comments for Title and Abstract:

  1. A general suggestion: I advise the authors to rewrite the Abstract in a way so that the relevant parts from the article are pointed out to really capture the realty important message of the present paper – think of writing the abstract by providing the relevant parts I right order (example 1 on lines 11-13: seem to be those that would normally be in the end of the abstract). In its current state, it is not serving the rest of the manuscript as good as it could. Keep the focus, and provide only what is in the focus.
  1. I advise the authors the rephrase the lines 11-12 (in current state it is way too packed and message is not clear): “Knowledge about individual risk factors for expected treatment requirements as the need of insulin at the time of diagnosis”.
  1. On lines 16-17: This is a good place to tell really clearly about the grouping (“study arms; 4 + the 33”) more clearly than it is now (see a comment below about this).
  1. What is the take home message based on this study? It has been stated more clear in other parts of the paper, yet it is really important to have in Abstract as well in a really clear format.

We thank the reviewer for this suggestion and rephrased the abstract substantially following the given advices.

Comments for Introduction: Introduction could use a bit more clear “red line” – I would advice the authors to cut some text that is taking the reader more far from the focus and to make it a more clear why and how the mental aspects of adhering to the insulin treatment is discussed, and how it supports the current study design where 5 groups are compared (4 + the 33, see a comment below)? Also, what are the general factors that predispose to GDM?

  1. A brief suggestion: I advise the authors to add a short but relevant sentence, where the risk of GDM in relation to pre-pregnancy diabetes is discussed just to add another angle to the story.

Dear reviewer, we agree and the link between the risk factor for GDM and the risk for the need of insulin during treatment is obvious and the risk factors for GDM are included in our model to predict medical treatment. In a former version of the manuscript we discussed this in brief, however we cut these parts since we feel it distract the reader from he here presented data which in deed analyzed the risk for insulin necessity rather than the risk of being diagnosed with GDM. Therefore, we hope to convince you to not discuss the risk to develop GDM in this particular paper. However, we changed the wording in the introduction to even more focus on the aim of the paper.

Comments for the other parts of the manuscript:

  1. I would ask the authors to discuss the results in the relation to pre-pregnancy diabetes a bit more in Discussion (to connect with the suggested sentence in Introduction).
  2. How was the final study population defined, is this full exclusion and inclusion criteria?
    1. Were there any pregnancies, where mothers have been previously diagnosed with any of the other diabetes sybtypes (T2D, T1D etc), and if yes, how were those individuals treated in this analysis (excluded/included?)? and 2. In case they were included, could an additional sensitivity analysis been done where those pregnancies are excluded?

Dear Reviewer, this is an interesting question. We only included patients with GDM in the primary analysis (n=990). Patients with any type of preexisting from of diabetes were not included.

    1. Were there differences among first-time pregnancies and others? If not done, a quick sensitivity analysis could be done.

Dear Reviewer, we only took the history of GDM in the analysis instead of the parity.

    1. Were there pregnancies, where a mother has been also previously diagnosed with GDM at any of the previous pregnancies?

Dear Reviewer, yes 7.7% of the cohort were women with a history of GDM as presented in Table 1.

    1. Were there differences between the different age groups? How much could age explain of these/potential differences related to treatment? I would advise the authors to discuss this more in Discussion.

Maternal age did not show an independent effect in the multivariate analysis. As you can also see in Table 1, there were no significant differences concerning the mean age in the different groups.

    1. Were any of the initial 990 mother-child pairs excluded due to prior metforming treatment? Was this also an exclusion criteria as it seems to have been stated as such on line 96?

None of the included or excluded patients was treated with metformin, since metformin treatment is currently restricted to exceptional cases according to the German guidelines.

  1. In this study, 487 mother-offspring pairs are included, what is the power? I would kindly ask the authors to perform the power calculations and add them as a supplementary table, if possible.

Dear Reviewer, since it was a retrospective study, we were only able to include the data that had been collected. We are well aware that -  in order to get unbiased estimates in a multivariate binary logistic regression model it is recommended that the sample should include at least 10 events per independent variable (according to Peduzzi P, Concato J, Kemper E, Holford TR, Feinstein AR. A simulation study of the number of events per variable in logistic regression analysis. J Clin Epidemiol. 1996;49(12):1373-9). In our primary analyses for the need for insulin more than the 170 necessary events were observed in the cohort, which is sufficient for the analyses of the 17 (including confounders) independent variables included in our multivariate analysis according to the rule mentioned above. This does not apply entirely for the subgroup regression analysis though. To lower the number of necessary events we could have excluded some of the confounders (gestational age, maternal age and parity) but we decided to stick with them due to their potential impact.  We did not discuss this issue explicitly in the discussion to avoid confusion of the reader that might not be familiar with this kind of approach. Please let us know if you think it is necessary to include this discussion in the text.

  1. I would kindly ask the authors to provide more information to the Figure 1, by additional boxes for treatment groups (this will help the reader to get the message faster), where this part of the text is visualized: both, a general insulin requirement and for the described therapeutic subgroups(Group comparisons of this study were based on four GDM treatment groups during pregnancy: ‘Diet’ (n=275), ‘Bolus’ (n=45), ‘multiple daily injections (MDI)’ (n=73) and ‘Basal’ (n=61)) within women diagnosed with gestational diabetes mellitus (GDM) after at 24 weeks of gestation or later”.

We thank you for that comment and changed Figure 1 by including the subgroups. We hope that it becomes more understandable for the reader now.

  1. Total study population is 487, and the N:s for four GDM treatment groups during pregnancy: ‘Diet’ (n=275), ‘Bolus’ (n=45), ‘multiple daily injections (MDI)’ (n=73) and ‘Basal’ (n=61) = 454. What is with the remaining 33 mother-child pairs? Are they a fifth study arm referred to as “a general insulin requirement”-group? I advice the authors to mention the “study arms” briefly in 2.1 Study Population, add “all 5 =4 treatment groups plus those 33 pairs?” groups to Figure 1, and mention them already in the Abstract to make this more clear to the readers from very beginning.

Dear Reviewer, this is absolutely correct and we have to apologize for this mistake. Due to including US parameters in our final analysis, we lost some further data sets due to incomplete data. Unfortunately, we did forget to change the passage in population section during the writing process. We changed it accordingly in Figure 1 and in the text. 

  1. In 2.3 Data collection, I would ask the authors to provide more information on population characteristics in a Table (Since Table 1 already exists, it would be great to add the reference to it already here).

We added a “see figure 1” at the end of this part.

  1. In Table 1, it is stated that total cohort N=454, however in Figure 1 and abstract, and some other parts in the text the number is 487. As advices above, make sure to explain what happened to the 33 pairs clearly from the beginning and to modify the Figure 1 accordingly. This will make it clear to the future readers as well.

See above. We apologize for the mistake again.

  1. I can see that ROC analyses were performed to evaluate the accuracy of the prediction of the certain treatment methods by using AUC with 95% confidence intervals. I would ask the authors to generate a supplementary table with the relevant AUC(ROC) results, and add a reference to the relevant Supplementary Table to line 122? In addition, were all variables modelled as continuous variables?

We decided to provide only the relevant AUCs in table 3. We could provide more AUCS in the supplements but we do not think that the reader could benefit from that data. To perform that AUC ROCS we only used continuous data. Category data was only included in the multivariate analysis. We added a reference to table 3 in the main text.

  1. In 2.4, were the analysis models adjusted for which factors? Add all relevant information to the results and other relevant parts.

All included variables are mentioned in this part: “Included variables were maternal age (years), gestational age at diagnosis, parity, prepregnancy weight (kg), prepregnancy BMI (kg/m2), systolic and diastolic blood pressure (mmHg), history of GDM, thyroid disorders, cardiovascular disorders, family history of diabetes, HbA1c at diagnosis (%) and fasting plasma glucose (FPG), 1h plasma glucose and 2h plasma glucose (all in mmol/l). Odds ratios (ORs) with 95% confidence interval (CI) are presented.” We did not adjust for further variables due to statistical limitations. We did include information about the included variables in Table 2.

  1. Could these risk factors be utilized for GDM-specific risk score generation? Further, how about the score, could it be used for disease prediction? An assessment of predictive performance of GDM-risk score for GDM-diagnosis would be of clinical and epidemiological relevance. This interesting topic could be briefly discussed in Discpreion (with relevant references).

Dear reviewer, as mentioned above we tried to focus on information retrieved during diagnosis of GDM to predict treatment requirements during pregnancy. We understand our data to be the initiation on thinking of and preforming research focusing on how patient characteristics should and could influence GDM management. And this independently of the classical GDM risk factors, we therefore avoided to mix this interesting topics. Nevertheless development of a prediction score leading to higher sensitivity and PPV by combining risk characteristics is obviously a good idea. However, we restrained from modelling during our analysis sticking to the aim to provide simple information which could be translated into daily practice. Especially by following your advices we feel to have met this goal now much better.

  1. Could some of these observed differences been explained by socioeconomic or environmental differences? This could be further discussed.

Social background, families eating habits, sleep quality and economic resources affect BMI, weight gain and blood pressure and thus the risk for insulin necessity during GDM pregnancy. However, this was not the focus of our study and due to the retrospective design we could not include this information in our analysis. Of course, for further studies with prospective design this should mandatorily be included. We added a short sentence into the discussion.

  1. I would advice the authors to start Discussion with summing up their own aim and results (keep the general rules about the chapter order in mind here as well) instead of the current first sentence.

We thank the reviewer for this comment and were happy to do so. The discussion has been reworded following your advices.

Other minor comments:

  1. Overall, readability is good, but in some places it could be improved.

We also aimed to impove readability.  

Reviewer 2 Report

The ability to predict which women diagnosed with gestational diabetes will require insulin therapy is a clinical question of clear interest to both clinicians and women. Weschenfelder and colleagues have attempted to address this question here, with the addition of predicting which insulin regimen (basal only, bolus only, or MDI) would be required. While analytical methods appear appropriate and results are clearly stated, more discussion is required regarding how, or indeed whether, the predictive cut-points identified are of use in clinical practice.

Major issues

P7, line 242 - NPV and PPV values for the FPG cut-off for need for any insulin therapy are low - so does this cut-off really have clinical utility? It is questionable what the utility of this would be, particularly when the cut-off for basal insulin is lower than the cut-off for any insulin. With cut-off FPG of 5.5 for need for any insulin, and 5.2 for basal insulin, what should clinicians advise women who have a FPG 5.3-5.4? These cutoffs would suggest they are unlikely to require insulin, but are at risk of requiring basal insulin, which is non-sensical.

Discussion:

The PPVs reported for bolus, basal and MDI insulin use are low – the authors should comment on the utility of using these cut-offs to predict insulin use. For each of these, the NPV is substantially higher, so arguably the clinical utility for these cut-points may be to reassure women whose levels are below these cut-points that their chance of requiring insulin is low. However, it is also important to consider that, for women who are initially counselled that their risk of requiring insulin is low, it may then be more traumatic to have created this expectation and then require insulin later who are expecting to not require insulin. It warrants discussion as to whether this risk of potential harm is worth the potential benefit that the reassurance of being stratified as low-risk for requiring insulin therapy may provide. The low PPVs would suggest these cut-offs are not useful for identifying women at high-risk of requiring these therapies.

Minor issues

P1, line 27 – suggest remove “primarily”, as the definition of GDM includes that it is diagnosed in pregnancy

P1, line 34 – suggest changing “insulin” to “medical”, as some regions/centres use other medical therapies prior to or with insulin (e.g. metformin, glibenclamide)

P2, line 61 – make explicit here that data collection was retrospective

P2, line 65 - What predictors commonly had data missing? This is a large number of potential cases excluded - was there anything different about excluded cases than included cases that may have introduced bias to the results?

P2, line 80 - Insulin regimens may change throughout pregnancy (e.g. commencing on basal only, with need to change to MDI as pregnancy progresses); does treatment group here refer to treatment at time of delivery?

P3, line 99 – Is there any data available on patient ethnicity?

P8, line 252 – The authors need to specify the comparator group for Asian women.

P8, line 259 – Suggest changing wording from “insulin patients” to “women requiring insulin therapy”.

P9, line 309 – An additional limitation is the inability to generalise findings to centres which offer the use of oral medical therapies prior to initiation of insulin.

P9, line 310 - However, HOMA and insulin levels are not used routinely in clinical practice, so I would argue it is reasonable that these were not included in the model.

Author Response

Dear Editor,

We would like to express our explicit thank the reviewers for careful reading our paper and providing such valuable comments. We are particular happy to respond to the comments provided and are convinced the manuscript has improved substantially.

Please find our point to point response to reviewer 2 on the following pages.

We deeply hope you will find the paper now acceptable to be published in your Journal.

Sincerely Tanja Groten

Reviewer 2

Comments and Suggestions for Authors

The ability to predict which women diagnosed with gestational diabetes will require insulin therapy is a clinical question of clear interest to both clinicians and women. Weschenfelder and colleagues have attempted to address this question here, with the addition of predicting which insulin regimen (basal only, bolus only, or MDI) would be required. While analytical methods appear appropriate and results are clearly stated, more discussion is required regarding how, or indeed whether, the predictive cut-points identified are of use in clinical practice.

We thank the reviewer for this appreciative comment and agree in the criticism of the discussion. The discussion was rephrased and we hope changes made are sufficient.

Major issues

P7, line 242 - NPV and PPV values for the FPG cut-off for need for any insulin therapy are low - so does this cut-off really have clinical utility? It is questionable what the utility of this would be, particularly when the cut-off for basal insulin is lower than the cut-off for any insulin. With cut-off FPG of 5.5 for need for any insulin, and 5.2 for basal insulin, what should clinicians advise women who have a FPG 5.3-5.4? These cutoffs would suggest they are unlikely to require insulin, but are at risk of requiring basal insulin, which is non-sensical.

We totally agree in this point and were happy to be enabled by the reviewer comments to now clearly express the results of the paper. The value here are the high NPV to rule out insulin treatment in 9 of 10 cases, if the cut off values are met. We now reworded main parts of the manuscript to emphasize this result.

Discussion:

The PPVs reported for bolus, basal and MDI insulin use are low – the authors should comment on the utility of using these cut-offs to predict insulin use. For each of these, the NPV is substantially higher, so arguably the clinical utility for these cut-points may be to reassure women whose levels are below these cut-points that their chance of requiring insulin is low. However, it is also important to consider that, for women who are initially counselled that their risk of requiring insulin is low, it may then be more traumatic to have created this expectation and then require insulin later who are expecting to not require insulin. It warrants discussion as to whether this risk of potential harm is worth the potential benefit that the reassurance of being stratified as low-risk for requiring insulin therapy may provide. The low PPVs would suggest these cut-offs are not useful for identifying women at high-risk of requiring these therapies.

We substantially reworded the discussion taking your valuable points made here into account.

Minor issues

P1, line 27 – suggest remove “primarily”, as the definition of GDM includes that it is diagnosed in pregnancy

This is correct. We changed it in the manuscript.

P1, line 34 – suggest changing “insulin” to “medical”, as some regions/centres use other medical therapies prior to or with insulin (e.g. metformin, glibenclamide)

Done

P2, line 61 – make explicit here that data collection was retrospective

Done

P2, line 65 - What predictors commonly had data missing? This is a large number of potential cases excluded - was there anything different about excluded cases than included cases that may have introduced bias to the results?

Since we decided to include a large number of variables in our regression model, the large number of cases that were excluded is caused by mostly one value missing in one case, leading to a large number summing up. Mostly information about blood pressure and fetal biometrics and oGTT values were the reason for exclusion.

On the other hand the characteristics of the initial entire cohort of 990 cases are comparable to this stud’s final cohort (maternal age 31 years, maternal BMI 26, 5 kg/m2, insulin therapy 42,2%, Bolus 11,6%; Basal 15,3%; MDI 15,5%, HbA1c 5,2% at diagnosis).We added this information to the limitation part of the discussion.

P2, line 80 - Insulin regimens may change throughout pregnancy (e.g. commencing on basal only, with need to change to MDI as pregnancy progresses); does treatment group here refer to treatment at time of delivery?

Dear Reviewer, thank you for that question. We changed the wording in the main text to make it clearer to the reader. Women were grouped according to the maximum therapy needed during pregnancy. Since we know that the insulin requirement often decreases towards the end of pregnancy and that the insulin therapy has to be stopped, the allocation to the groups would otherwise be incorrect if we were to refer only to the time of birth. We added a comment in the text.  

P3, line 99 – Is there any data available on patient ethnicity? P8, line 252 – The authors need to specify the comparator group for Asian women.

Thank you for this question. For ethnicity it has to be stated that in the Jena obstetric population the proportion of non-Caucasian women is far below 10%. We added that information and discuss it in the main text.  

P8, line 259 – Suggest changing wording from “insulin patients” to “women requiring insulin therapy”.

We changed the wording in the text.

P9, line 309 – An additional limitation is the inability to generalise findings to centres which offer the use of oral medical therapies prior to initiation of insulin.

We added this valuable comment in the limitations section. Thank you for pointing that out.

P9, line 310 - However, HOMA and insulin levels are not used routinely in clinical practice, so I would argue it is reasonable that these were not included in the model.

Thank you for agreeing with us. We added a comment in the limitations section.

Round 2

Reviewer 1 Report

General comments - 2nd round of peer-review:

The authors have addressed and corrected the manuscript according to the previous review round. I find, that this manuscript has improved significantly since the prior revision and taking into account comments from both reviewers. However, I did still find few things to be worth raising follow-up questions or suggestions, and hope that the authors benefit from these as well. I also find, that readability could still be improved and it can be achieved by English editing, that is a common practice – I raise this, because I find it really important to deliver the wanted message (which in some parts either gets out of intended focus or misunderstood).

For Abstract:

  1. L11-12: I suggest that authors remove the last part of the sentence, because it is now making the sentence more confusing compared to the previous version. I would also rephrase the sentence as follows: “Receiving the diagnosis of gestational diabetes is distressing for most pregnant women, fearing insulin treatment might be necessary”. I would do this rephrasing only if it is true (this is, because I assume it was what was intended): In its original phrasing, this sentence meant that the “diagnosis of GDM is mostly distressing and nothing more”, and _not_ that the “GDM diagnosis is distressing for the most pregnant women”. So, I advise the authors to keep the phrasing that is intended.
  2. L18-19: I advise the authors to add the N:s for the arms. “…further stratified treatment options: diet (N=275), bolus insulin only (N=45), basal insulin only (N=73) and multiple daily injections (N=61).”
  3. L21-22: “Multivariate analysis proofed this relevance of heterogeneity.” I would either rephrase this sentence and support it with some relevant data, or remove it completely from the abstract.
  4. L20: “Treatment groups differed significantly…” I would advise the authors to add the cut-off value for significance that was used (e.g. p<0.05), with the actual results as numbers and CI 95% values for the variables that are mentioned.

For Introduction:

  1. Overall, the text has improved a lot in Introduction. It served the focus of the paper much better in its current state.
  2. However, the text could still be rephrased a bit, to chop heavy sentences to shorter and more focused ones. Some parts could be divided with commas. At its current state, the sentences are way too heavy and for some sentences their meaning changes completely because no commas are placed to relevant places. For example, the first sentence (L29-22) would be much more reader friendly and accurate by adding commas around the “,with a worldwide incidence of up to 11%,”.
  3. L33, order of the sentence is not accurate and it´s missing a reference. I advise the authors to flip the “Treatment of GDM according to German S3 Guidelines” to “According to German S3 Guidelines (add REF here to the German guidelines), treatment of GDM is based on…”.
  4. L34: Just a minor issue with wording: I would advise the authors to change “dietary schooling” into dietary education/advising/tutoring/coucelling. This is, because “schooling” refers to degree-oriented education given at schools.
  5. I would rephrase the L37-39 as follows, to make it more focused: “According to German guidelines, metformin treatment is currently restricted to exceptional cases and not recommended. In Germany about 30% of the GDM patients receive insulin treatment during pregnancy instead.

For other parts of the manuscript:

  1. I thank the authors for all the explanations and corrections. I have a follow-up question related to the women with previous GDM diagnosis: For the 7.7% of the women with history of GDM; were there any differences between the women with history GDM and those with not - in general? How much could a previous GDM explain of these potential differences related to treatments? I would add this part to the discussion as well, if relevant.
  2. For the question related to the power calculations: I thank the authors for an explanation and do think that it would be good to add a sentence about the power calculations to the methods section, with a reference to a “supplementary information” (or such) more profound explanation in Supplementary Material. Also a line about the power (especially explaining some of the results in smallest strata), could be added to the limitations part in Discussion.
  3. I would add the thresholds for P-values (e.g. P<0.001) that have been used to define significance to Table 1 (to the table legend).
  4. Professional English editing could be used throughout the text, to made sure the right message is delivered to the reader. This is a minor thing, but important (these improvements are similar to the suggestions provided for Abstract and Introduction parts). 

General comments for Supplementary Material - 2nd round of peer-review:

  1. In Supplementary Material for Table 4: As a general rule, tables and figures should be self-explanatory, meaning that they should be understandable without the need to glance to the main text. They should also be consistent within, and with other similar tables. This coherence should also then be reaching out to the main text. This minor “styling” makes it easier to the reader to compare the values and find the relevant information quickly.
    1. All abbreviations (e.g. NICU, Neonatal intensive care unit; LGA, Large for gestation age; etc….) should be written out under the table in alphabetical order.
    2. A minor suggestion: If preferred, remove the % from the cells with actual values, and have it in “the variable name above the each relevant column (e.g. Insulin (%)), it would make the table neater – however, this is a matter that is highly about the preferences, and therefore, it is good as it is now as well.
    3. Add (N) to “Data available” (e.g. Data available (N)).
    4. For “Diet control” and other like that, add also information on “out of total population (N=X)” or something else, depending on the variable to have also information on how the % has been calculated.
    5. Decimals: make sure the number of decimals is equal throughout the table (e.g. 1.00 and 0.23, instead of 1 and .23) , and results as well in main manuscripts. Add 0 to all cells, where starting the values with “.”, (e.g. 0.618 instead of .618).
    6. Consistency: “hypoglycemia” should start with a capital letter as well, if all other variable names already are.

Author Response

Dear Reviewer, we again thank you for the valuable and appreciating comments, which we are happy to respond to.

The authors have addressed and corrected the manuscript according to the previous review round. I find, that this manuscript has improved significantly since the prior revision and taking into account comments from both reviewers. However, I did still find few things to be worth raising follow-up questions or suggestions, and hope that the authors benefit from these as well. I also find, that readability could still be improved and it can be achieved by English editing, that is a common practice – I raise this, because I find it really important to deliver the wanted message (which in some parts either gets out of intended focus or misunderstood).

We did perform a further English editing taking into account to focus on the message to be delivered.

For Abstract:

  1. L11-12: I suggest that authors remove the last part of the sentence, because it is now making the sentence more confusing compared to the previous version. I would also rephrase the sentence as follows: “Receiving the diagnosis of gestational diabetes is distressing for most pregnant women, fearing insulin treatment might be necessary”. I would do this rephrasing only if it is true (this is, because I assume it was what was intended): In its original phrasing, this sentence meant that the “diagnosis of GDM is mostly distressing and nothing more”, and _not_ that the “GDM diagnosis is distressing for the most pregnant women”. So, I advise the authors to keep the phrasing that is intended.
  2. L18-19: I advise the authors to add the N:s for the arms. “…further stratified treatment options: diet (N=275), bolus insulin only (N=45), basal insulin only (N=73) and multiple daily injections (N=61).”
  3. L21-22: “Multivariate analysis proofed this relevance of heterogeneity.” I would either rephrase this sentence and support it with some relevant data, or remove it completely from the abstract.
  4. L20: “Treatment groups differed significantly…” I would advise the authors to add the cut-off value for significance that was used (e.g. p<0.05), with the actual results as numbers and CI 95% values for the variables that are mentioned.

Dear reviewer, the problem with the abstract is, that length is restricted to 200 words. However, we changed the first sentence to “The diagnosis of gestational diabetes is usually very stressful for pregnant women, especially because they fear that insulin treatment may become necessary.” (Instead of “Receiving the diagnosis of gestational diabetes is mostly distressing for pregnant women, fearing insulin treatment might be necessary”). The point and aim we would like to state with this sentence is that women fear the diagnosis of GDM due to the threat/fact that insulin therapy becomes necessary. We hope this becomes more clear form the sentence now. For that reason we could not extend our result explanation in the abstract but we added the p <0.05 to table 3 to recall the reador of the chosen statistical value. However, we followed your suggestions and removed the sentence about heterogeneity, which gave us the possibility to follow your suggestions and sill stay within the 200-word limit.

For Introduction:

  1. Overall, the text has improved a lot in Introduction. It served the focus of the paper much better in its current state.

Thank you again for your helpful comments!

  1. However, the text could still be rephrased a bit, to chop heavy sentences to shorter and more focused ones. Some parts could be divided with commas. At its current state, the sentences are way too heavy and for some sentences their meaning changes completely because no commas are placed to relevant places. For example, the first sentence (L29-22) would be much more reader friendly and accurate by adding commas around the “,with a worldwide incidence of up to 11%,”.
  2. L33, order of the sentence is not accurate and it´s missing a reference. I advise the authors to flip the “Treatment of GDM according to German S3 Guidelines” to “According to German S3 Guidelines (add REF here to the German guidelines), treatment of GDM is based on…”.
  3. L34: Just a minor issue with wording: I would advise the authors to change “dietary schooling” into dietary education/advising/tutoring/coucelling. This is, because “schooling” refers to degree-oriented education given at schools.
  4. I would rephrase the L37-39 as follows, to make it more focused: “According to German guidelines, metformin treatment is currently restricted to exceptional cases and not recommended. In Germany about 30% of the GDM patients receive insulin treatment during pregnancy instead.

We changes the text in the introduction according to your advice and did some editing to further improve understandability.

For other parts of the manuscript:

  1. I thank the authors for all the explanations and corrections. I have a follow-up question related to the women with previous GDM diagnosis: For the 7.7% of the women with history of GDM; were there any differences between the women with history GDM and those with not - in general? How much could a previous GDM explain of these potential differences related to treatments? I would add this part to the discussion as well, if relevant.

Dear reviewer, we understand this point and we were actually ourselves surprised and disappointed when realizing that the fact of history of GDM did not reveal to be relevant for the need of insulin. This was shown by our multivariate analysis. However, there were only 35 women with a consecutive diagnosis. Furthermore, to our experience some women improve their health status following the first GDM-pregnancy while other do not. This could explain, why the just the fact of having had GDM does not impact on the course of the second pregnancy. Secondly, information about whether patients with a history of GDM had insulin treatment during that pregnancy was not documented. We added some sentences to the discussion, however avoiding to go further into this discussion.

  1. For the question related to the power calculations: I thank the authors for an explanation and do think that it would be good to add a sentence about the power calculations to the methods section, with a reference to a “supplementary information” (or such) more profound explanation in Supplementary Material. Also a line about the power (especially explaining some of the results in smallest strata), could be added to the limitations part in Discussion.

We added some text according to you suggestions. Thank you.

  1. I would add the thresholds for P-values (e.g. P<0.001) that have been used to define significance to Table 1 (to the table legend).

As our significance level are mentioned (< 0.05) for all statistical analysis in the method section. However, we added this informations to all tables.

  1. Professional English editing could be used throughout the text, to made sure the right message is delivered to the reader. This is a minor thing, but important (these improvements are similar to the suggestions provided for Abstract and Introduction parts).

English editing was performed. However, for external professional service the 3-days deadline was not sufficient

General comments for Supplementary Material - 2nd round of peer-review:

  1. In Supplementary Material for Table 4: As a general rule, tables and figures should be self-explanatory, meaning that they should be understandable without the need to glance to the main text. They should also be consistent within, and with other similar tables. This coherence should also then be reaching out to the main text. This minor “styling” makes it easier to the reader to compare the values and find the relevant information quickly.
    1. All abbreviations (e.g. NICU, Neonatal intensive care unit; LGA, Large for gestation age; etc….) should be written out under the table in alphabetical order.
    2. A minor suggestion: If preferred, remove the % from the cells with actual values, and have it in “the variable name above the each relevant column (e.g. Insulin (%)), it would make the table neater – however, this is a matter that is highly about the preferences, and therefore, it is good as it is now as well.
    3. Add (N) to “Data available” (e.g. Data available (N)).
    4. For “Diet control” and other like that, add also information on “out of total population (N=X)” or something else, depending on the variable to have also information on how the % has been calculated.
    5. Decimals: make sure the number of decimals is equal throughout the table (e.g. 1.00 and 0.23, instead of 1 and .23) , and results as well in main manuscripts. Add 0 to all cells, where starting the values with “.”, (e.g. 0.618 instead of .618).
    6. Consistency: “hypoglycemia” should start with a capital letter as well, if all other variable names already are.

This was done and we thank the reviewer to remind us of these obvious rules.
